# Exercise-Based Interventions in Hemodialysis Patients: A Systematic Review with a Meta-Analysis of Randomized Controlled Trials

**DOI:** 10.3390/jcm9010043

**Published:** 2019-12-24

**Authors:** Špela Bogataj, Maja Pajek, Jernej Pajek, Jadranka Buturović Ponikvar, Armin H. Paravlic

**Affiliations:** 1Department of Nephrology, Ljubljana University Medical Centre, Ljubljana 1000, Slovenia; sspelabogataj@gmail.com (Š.B.); jernej.pajek@mf.uni-lj.si (J.P.); jadranka.buturovic@kclj.si (J.B.P.); 2Faculty of Sport, University of Ljubljana, Ljubljana 1000, Slovenia; maja.pajek@fsp.uni-lj.si; 3Faculty of Medicine, University of Ljubljana, Ljubljana 1000, Slovenia; 4Science and Research Centre, Institute for Kinesiology Research, Koper 6000, Slovenia

**Keywords:** exercise, hemodialysis, physical performance, functional capacity, oxygen consumption, inflammation, meta-analysis

## Abstract

There is a lack of agreement on the efficacy of different exercise interventions in hemodialysis patients. We analyzed which exercise type is the most beneficial in terms of functional fitness and inflammation. A literature search of seven databases yielded 33 studies that met the inclusion criteria. Compared with an inactive control, the intervention group showed moderate effects (ES = 0.74; 95% CI 0.35 to 1.14; *p* < 0.001; and ES = 0.70; 95% CI 0.39 to 1.01; *p* < 0.001; respectively) on functional capacity (six-minute walk test) and oxygen consumption. Small nonsignificant effects were observed for aerobic (ES = −0.36; 95% CI −0.85 to 0.13; *p* = 0.154) and resistance (ES = −0.44; 95% CI −1.07 to 0.19; *p* = 0.169) training types, whereas moderate effects were found for combined (ES = −0.69; 95% CI −1.47 to 0.10; *p* = 0.088) training type based on a 10-repetition sit-to-stand test. Further, large and small effects were observed for aerobic (ES = −1.21; 95% CI −1.94 to −0.49; *p* = 0.001) and resistance training (ES = −0.54; 95% CI −0.90 to −0.17; *p* = 0.004) types on C-reactive protein. Overall, the results showed the numerically largest effect sizes for combined types compared to solely aerobic or resistance training types, with the differences between training types not reaching statistical significance. There was a significant modifying impact of age, training frequency, and session duration on performance and inflammatory outcomes.

## 1. Introduction

The global prevalence of chronic kidney disease (CKD) is estimated to be at an all-time high [1]. End-stage kidney disease (ESKD) needs to be treated with renal replacement therapy [2]. Hemodialysis (HD) is the most widely used renal replacement treatment for ESKD [3]. Patients undergoing HD are mostly physically inactive [4] and have reduced functional capacities compared to healthy individuals [5], which contributes to a decreased quality of life and consequently increases the risk of mortality [6].

The decline in muscle strength is larger in HD patients compared to the age-matched general population, which could be, in part, a consequence of the increased incidence of intermittent hospitalizations [7]. Generally, the state of chronic residual uremic syndrome leads to physical inactivity, which further has a negative consequence on overall health status. Additionally, HD patients have a significantly increased risk of vascular and cardiac disease [8]. Cardiovascular abnormalities and anemia cause a reduced oxygen supply, which limits the patient’s aerobic capacity [9] to only 50% that of the general population (aerobic capacity has a vital role in maintaining the activities of daily living) [10]. Another additional problem in HD patients is inflammation, which is indicated by elevated C-reactive protein (CRP) and other inflammatory markers, and can be used as a predictor of forthcoming cardiovascular morbidity and mortality [11].

Physical activity has been shown to have various positive effects on HD patients [12,13]. Exercise is also crucial for retaining physical independence [14]. Some of the important benefits linked to exercise include an improvement in physical fitness [15], aerobic capacity [16], dialysis adequacy (measured as urea Kt/V) [12], quality of life [17], and reduced depressive symptoms [18]. Exercise training can be delivered over different timings and modalities. For convenience, exercise is often delivered on dialysis days at a dialysis center (before or during the dialysis procedure). Intradialytic exercise on a customized bicycle has been identified as the most prevalent form of organized exercise among HD patients, presumably because it does not involve any extra time, and the patient is supervised by medical staff while exercising [12,19]. However, the patient’s movements are limited to a lying or sitting position. Other reported modalities of exercise training in studies with HD patients include resistance training, aerobic training out of the dialysis procedure, and combined resistance and aerobic training. Since there are many studies already published on the topic, it would be valuable to determine which modality is actually the most beneficial for a patient’s physical performance and overall benefit.

A recent meta-analysis [16] shed light on the effects of different exercise modalities. The authors stated that combined training is the most effective modality for increasing maximal oxygen consumption. Another meta-analysis [20] reported different results, showing that the maximal oxygen consumption was significantly increased by aerobic and resistance training without significant effects from combined training. Nevertheless, they included only two studies that reported the effect of combined training. This lack of agreement on the efficacy of different exercise interventions could be attributed to the differences in the study inclusion criteria or differences in the duration, intensity, delivery type, and frequency of exercise. Accordingly, an evidence-based answer to the uncertainty over the most beneficial type of exercise for dialysis patients is needed. Thus, we conducted this meta-analysis to answer this question in a more inclusive way with a more complete inclusion of relevant studies than previous meta-analyses.

We aimed to provide a broad, systematic, and evidence-based assessment that will update the recommendations of exercise interventions for the HD population. Our goal was to precisely identify which type of exercise modality (i.e., aerobic, resistance, or combined exercise) is the most beneficial for HD patients. In addition, a meta-regression analysis was performed to evaluate the impact of significant, potentially mediating covariates that could influence the intervention’s effects on measures of functional fitness/capacities and inflammation biomarkers.

## 2. Materials and Methods

### 2.1. Search Strategy

Two authors (Š.B. and A.H.P.) developed the search strategy used in this study. Literature searches of the PubMed, DOAJ, Web of Science, PEDro, KoreaMed, Google Scholar, and ScienceDirect databases were conducted from February 2018 to March 2019. We searched electronic databases using the following keywords: “hemodialysis”, “haemodialysis” “exercise”, “physical rehabilitation”, “chronic kidney disease”, “renal failure”, “renal replacement therapy”, “nephrology”, “dialysis”, “6-min walk test”, “VO2max”, “oxygen consumption”, “CRP”, “sit-to-stand test”, “aerobic capacity”, “physical activity”, “training”, “exercise training”, “aerobic training”, “combined training”, “strength training”, or “resistance training”. The references of potentially relevant articles were scanned to identify additional applicable studies. To clarify the results and methods sections, we contacted the authors through e-mail or via the ResearchGate platform. No language restrictions were encountered. For our analysis, we selected controlled randomized trials that determined the effects of different exercise modalities on functional capacity (a 6-min walk test), oxygen consumption (peak/maximum VO_2_), lower limb strength endurance (10 repetition sit-to-stand test), and inflammation (C-reactive protein).

### 2.2. Inclusion and Exclusion Criteria

Eligible studies that passed selection according to the determined inclusion and exclusion criteria were included. The inclusion criteria were the following: adult end-stage kidney disease HD patients, randomized controlled studies that included aerobic, resistance, or combined exercise training with a pre–post intervention that lasted more than eight weeks, studies that evaluated the effects of exercise on at least one of the outcomes in this meta-analysis. Criteria for exclusion were non-HD patients and young (<18 years) HD patients, animal studies, systematic reviews and meta-analyses, conference abstracts, theses, and case reports. Our primary endpoint was the functional capacity measured by a six-minute walk test. Secondary endpoints were oxygen consumption, 10 repetition sit-to-stand test, and C-reactive protein.

### 2.3. Screening Strategy

Two authors independently (Š.B. and A.H.P.) performed the literature search, along with study identification, screening, quality assessment, and data extraction. The titles were initially screened by the reviewers during their electronic searches. All papers beyond the scope of this meta-analysis were excluded. Secondly, the abstracts were assessed using pre-agreed inclusion and exclusion criteria. Finally, the full texts of the remaining papers that met the inclusion criteria were reviewed to make a final decision on their inclusion in the meta-analysis. Any disagreements between the reviewers were resolved through a third reviewer (J.P.). If the full text of any paper was not available, the corresponding author was contacted by mail or via the ResearchGate platform. The study selection process, as described above, is illustrated in Figure 1 following PRISMA (Preferred Reporting Items for Systematic Reviews and Meta-analyses) guidelines [21].

### 2.4. Data Extraction

The Cochrane Consumers and Communication Review Group’s data extraction protocol was used to extract participant information, including sex, age, sample size, training status, description of the intervention (duration of the intervention; training frequency, total volume of training (per whole study, per week)), single session duration, type of exercise intervention (aerobic, resistance, combined training), type of comparison group (passive or active control)), study design, and study outcomes [22].

### 2.5. Quality Assessment

Two authors (Š.B. and M.P.) independently assessed the quality and risk of bias using checklists. Agreement between the two reviewers was assessed using *k* statistics for full-text screening and a rating of relevance and risk of bias. In the case of a disagreement on the risk of bias, a third reviewer made the final decision. The Physiotherapy Evidence Database (PEDro) scale was used to assess the methodological quality of the included studies [23]. The quality assessment score was interpreted using the following 10-point scale: ≤3 points was considered as poor quality, 4–5 points as moderate quality, and 6–10 points as high quality. The PEDro scale consists of 11 items designed to rate the methodological quality. Each satisfied item contributes 1 point to the overall PEDro score (with a range of 0–10 points).

### 2.6. Completeness of Intervention Description

The completeness of the intervention description for both the intervention and control groups was assessed using the 12-item TIDieR checklist [24]. This checklist includes the short name of the intervention: why (the rationale); what (the materials); what (the procedure); who (provided the intervention); how (modes of delivery); where, when, and how much; tailoring; modifications; how well (planned outcomes); and how well (actual outcomes).

### 2.7. Statistical Analysis

The meta-analyses were performed using the Comprehensive Meta-analysis software (Version 2.0, Biostat Inc., Englewood, NJ, USA). The effect size was calculated according to the following formula: $\mathrm{ES}=\frac{M_{\mathrm{post}}-M_{\mathrm{pre}}}{{\mathrm{SD}}_{\mathrm{pooled}}}$, where ES represents the effects size, M_post_ is the mean value after treatment (POST), M_pre_ is mean value before treatment (PRE), and SD_pooled_ represents the pooled standard deviation (SD). We calculated the mean differences and 95% confidence intervals (CIs) for the included studies. The I^2^ measure of inconsistency was used to assess between-study variability, whereby values of 25%, 50%, and 75% represent low, moderate, and high statistical heterogeneity, respectively [25]. Although the heterogeneity of the effects in this meta-analysis ranged from 0% to 78.9%, we decided to apply a random-effects model in all comparisons to determine the pooled effect of physical exercise interventions on dependent measures of interest [26]. Furthermore, a random-effects meta-regression was performed to examine whether the effects of physical exercise interventions on dependent measures of interests were moderated by patients related (i.e., a patient’s age and time on dialysis) and different training variables, respectively. The training variables included the following: duration of intervention, weekly training frequency, duration of a single training session, the total number of training sessions, and total training volume (min).

The magnitude of the effect was interpreted using the following criteria: trivial (<0.20), small (0.21–0.60), moderate (0.61–1.20), large (1.21–2.00), very large (2.01–4.00), and extremely large (>4.00) [27]. A significance level of *p* ≤ 0.05 was used for all analyses. The negative direction of ES indicates a better effect from an intervention on C-reactive protein and the 10 repetition sit-to-stand test.

## 3. Results

### 3.1. Study Characteristics

A total of 2807 papers were identified across the databases in the initial search, and an additional two papers were selected through other sources. After duplicates were removed, 2204 articles remained. After a screening of the title and abstract, 2100 articles were discarded. The full texts of the 104 remaining papers were assessed in more detail for eligibility. Each paper was carefully read and coded for study characteristics, participant information, description of the training intervention, and study outcomes. Seventy-one papers did not meet the inclusion criteria, while 33 papers that met the inclusion criteria were included in the meta-analysis and systematic review with a total of 1274 patients (Figure 1). All eligible studies were randomized controlled trials published from 1995 to 2018.

The minimum sample size was 14, and the maximum size was 96. The average age of all included subjects was 55.7 ± 9.6 years. The duration of interventions ranged from 8 to 40 weeks. The most common intervention period was 12 weeks [28,29,30,31,32,33,34,35,36,37,38]. The most frequent exercise type was aerobic exercise. Combined [35,37,39,40,41,42,43,44,45,46] and resistance exercise [33,34,36,38,47,48] were the next most common. Three times per week was the most popular exercise frequency, followed by two times per week [37] and four [44] times per week. In the majority of cases, the intervention was performed during the dialysis procedure. In the included interventions, the intensity of exercise often was moderate (13–14 on a Borg scale, 65% of VO_2max_, 55%–60% of the peak power, 65% of the 1RM). Moreover, a single exercise duration ranged from 13 min [28] to 90 min [49]. Detailed characteristics of the included interventions are shown in Table 1 and Appendix A.

### 3.2. Quality Assessment

Two authors of this article (Š.B. and M.P.) performed the search, coding, and appraisal of methodological quality independently, with discussion and consensus over any observed differences. Cohen’s kappa was 0.84 (95% CI 0.73 to 0.95), indicating an excellent inter-rater agreement. PEDro scores ranged from 3 to 9 out of 10, with a median score of 6.0 (Table 2). The *k* agreement score between the reviewers was *k* = 0.91. Nine trials blinded the assessor, fifteen used intention-to-treat analysis, and twelve concealed allocation.

### 3.3. Completeness of Intervention Description

The experimental group conditions were more completely reported compared than the control group conditions. The percentage of studies satisfying each TIDieR item ranged from 17% to 92% for the experimental group (Figure 2) compared to 0% to 50% for the control group (Figure 3). The procedures (TIDieR item 4, T4) used in the intervention and details about the exercise program (T8) were the most satisfactorily reported, while information on how the intervention was modified (T10) or monitored irrespective of the group (items 11 and 12) was poorly reported. Adequate details about the rationale of the intervention (T2), intervention provider (T5), materials (T3), how the intervention was tailored (T9), and modes of delivery (T6) were provided for the experimental group in more than 67% of trials, whereas the control group achieved less than 25%.

### 3.4. Effects of the Interventions on Measures of Functional Fitness/Capacities

#### 3.4.1. Primary Outcome: Changes in Functional Capacity Measured by a 6-min Walk Test

The summarized effects of 22 ESs showed a small effect (ES = 0.44; 95% CI 0.24 to 0.65; *p* < 0.001) on a 6-min walk test (Figure 4). The statistical heterogeneity was moderate (*I*^2^ = 49.60%; *p* = 0.006); therefore, a sub-analysis and meta-regression analysis were performed. A sub-group analysis revealed that the effects of the intervention were significantly moderated by the conditioning assignment of the control group (Q = 6.15; *p* = 0.013). Hence, small and moderate positive effects were observed when the intervention was compared with the active (ES = 0.18; 95% CI −0.03 to 0.39; *p* = 0.098) and inactive (ES = 0.74; 95% CI 0.35 to 1.14; *p* < 0.001) control groups, respectively. When the type of exercise was compared, a trend towards a significant moderating effect was observed (Q = 4.73; *p* = 0.094). Hence, a small effect was observed for aerobic types of exercise (ES = 0.48; 95% CI 0.24 to 0.73; *p* < 0.001), while moderate and trivial effects were observed for combined (ES = 0.71; 95% CI −0.06 to 1.48; *p* = 0.072) and resistance training (ES = 0.10; 95% CI −0.19 to 0.39; *p* = 0.498) exercise types, respectively. In addition, regression analysis indicated that only the age of patients (*p* = 0.025) and weekly frequency of training (*p* < 0.001) significantly predicted the effects of the intervention on a 6-min walk test (Table 3).

#### 3.4.2. Secondary Outcomes

##### Changes in Oxygen Consumption Measured by VO_2max_ and _peak_VO_2_ Tests

The summarized effects of 20 ESs showed moderate effects (ES = 0.58; 95% CI 0.32 to 0.85; *p* < 0.001) on oxygen consumption (Figure 5). The statistical heterogeneity was moderate (I^2^ = 57.4%; *p* = 0.001); therefore, a sub-analysis and meta-regression analysis were performed. The sub-group analysis revealed that the effects of the intervention were significantly moderated by the conditioning assignment of the control group (Q = 4.12; *p* = 0.042). Hence, small and moderate effects were observed when the intervention was compared with the active (ES = 0.23; 95% CI −0.11 to 0.56; *p* < 0.001) and inactive (ES = 0.70; 95% CI 0.39 to 1.01; *p* < 0.001) control groups, respectively. In addition, when the type of exercise was assessed as a moderating factor, a not significant difference was shown (Q = 0.41; *p* = 0.522). Thus, small and moderate effects were found for aerobic (ES = 0.50; 95% CI 0.08 to 0.91; *p* = 0.019) and combined (ES = 0.67; 95% CI 0.33 to 1.01; *p* < 0.001) types of exercise, respectively. In addition, regression analysis indicated that only the duration of a single training session (*p* = 0.025) significantly predicted the effects of the intervention on oxygen consumption, while time on dialysis (*p* = 0.078) and total training volume (*p* = 0.084) showed a trend (Table 3).

##### Changes in Lower Limb Strength Endurance Assessed by 10 Repetition Sit-to-Stand Test

The summarized effects of six ESs showed a moderate effect (ES = −0.56; 95% CI −0.94 to −0.18; *p* = 0.004) on a 10 repetition sit-to-stand test (Figure 6). The statistical heterogeneity was moderate (I^2^ = 71.6%; *p* = 0.003), and therefore a sub-analysis was performed. Sub-group analysis revealed that the effects of the intervention were moderated via a conditioning assignment of the control group. However, this difference was not significant (Q = 2.34; *p* = 0.126). Also, when the type of exercise was assessed as a moderating factor, a not significant difference was shown (Q = 0.48; *p* = 0.786). Thus, small effects were observed for aerobic (ES = −0.36; 95% CI −0.85 to 0.13; *p* = 0.154) and resistance (ES = −0.44; 95% CI −1.07 to 0.19; *p* = 0.169) types of exercise, whereas moderate effects were found for combined (ES = −0.69; 95% CI −1.47 to 0.10; *p* = 0.088) types of exercise, respectively. A meta-regression analysis was not performed because fewer studies were included.

### 3.5. Effects of the Interventions on the Measures of Inflammation Biomarkers

#### Changes in the C-Reactive Protein (CRP)

The summarized effects of the 14 ESs showed a moderate effect (ES = −1.03; 95% CI −1.52 to −0.53; *p* < 0.001) on the CRP (Figure 7). The statistical heterogeneity was high (I^2^ = 78.1%; *p* < 0.001) and, therefore, a sub-analysis and meta-regression analysis were performed. A sub-group analysis revealed that the effects of the intervention were moderated by the conditioning assignment of the control group. However, this difference was not significant (Q = 0.23; *p* = 0.628). Also, when the type of exercise was assessed as a moderating factor, a not significant difference was shown (Q = 2.67; *p* = 0.102), where large and small effects were found for aerobic (ES = −1.21; 95% CI −1.94 to −0.49; *p* = 0.001) and resistance training (ES = −0.54; 95% CI −0.90 to −0.17; *p* = 0.004) types of exercise, respectively. We found no studies investigating the effects of combined training on CRP. In addition, a regression analysis indicated that only the age of patients (*p* = 0.001) and time on dialysis (*p* = 0.035) significantly predicted the effects of an intervention on CRP (Table 3).

## 4. Discussion

The presented meta-analysis identified a total of 33 randomized controlled trials investigating the effects of exercise training in HD patients. The aim of this study was to define the effects of different exercise modalities on functional capacity (6-min walk test), oxygen consumption (VO_2max_ and _peak_VO_2_), lower limb muscle strength endurance (10 repetition sit-to-stand test), and inflammation, as reflected by CRP. The summary of the results of our analysis indicates that all selected outcomes improved with exercise training. In this way, our findings are consistent with past reviews [13,61,62]. Our analysis, however, differs from previous studies in that we performed a meta-regression analysis of various variables and a subgroup analysis of different exercise modalities to explore their impacts. Here, combined training provided the largest beneficial effect sizes with patient age, training session frequency, and duration being the possible modifying parameters.

As a primary endpoint, we chose the 6-min walk test, since this is a useful test to stage disease severity, provides treatment efficacy information, and overall represents a functional test associated with the activities of daily living [63]. We found improvements in functional capacity measured by the 6-min walk test following aerobic and combined types of exercise interventions. These results are consistent with those of the previous review [61], where a meta-analysis of seven trials among HD patients revealed a significant positive impact on functional capacity, although these trials did not examine the effects of training modalities. Our analysis showed that aerobic training is the most convincingly beneficial modality for improving 6-min walk results among HD patients. However, the largest effect size was found for combined training modality. Two trials [29,54] specifically used walking as an aerobic intervention, so the results might be partly attributed to walking practice being identical to the tests that were performed. Meta-regression showed that age and weekly training frequency are significant predictors for intervention effects. Most of the studies included here (90%; 19 out of 21) prescribed a training frequency of three times weekly, including studies with the largest improvements in functional capacity.

A limitation of the 6-min walk test is that the results can be affected by the patient’s height and body fat [64]. Therefore, besides this test, we examined the outcome of oxygen consumption at maximal exertion (VO_2max_) and _peak_VO_2_ at submaximal exertion. These parameters are considered to be significant predictors of survival among the dialysis population [6]. Our meta-analysis found that exercise increased maximal oxygen consumption compared to the control, and the sub-group analysis revealed beneficial effects for both the aerobic and combined training modalities. These findings match those of a recent meta-analysis [16], which showed that both aerobic and combined training significantly improved VO_2max_. In our analysis, combined training had a larger effect than aerobic training, which may be due to an increase in basal metabolism [65]. The meta-regression showed that the duration of a single training session predicted the intervention’s effect. Durations of sessions in the included studies ranged from 30 to 90 min. The highest impact was seen in the three studies with an exercise duration of 50 min per session [39,44,46].

One of the most frequent physical tests for lower extremity strength widely used in clinical settings is the sit-to-stand test in various forms (five repetitions, 10 repetitions, and 60 or 30 s test). However, a 10-repetition variant is considered to be the most sensitive tool when measuring the effect of intra-dialytic training interventions [48]. A 10 repetition sit-to-stand test measures the time required to complete ten full stands from a sitting position. This test indicates lower limb strength endurance and is well associated with the activities of daily living [66]. The scores for ten repetition sit-to-stand test were considered to be a strong predictor of mortality among patients with chronic obstructive pulmonary disease [67]. Notably, our meta-analysis was the first to gather all studies that examined the effects of exercise on this test among HD patients. Our results showed that HD patients could improve their performance in 10 repetition sit-to-stand test with no significant difference among the exercise modalities used. However, the most prominent effect was found for combined training.

In dialysis patients, there is an ongoing low-intensity chronic inflammation reflected by elevated CRP, which represents a substantial contributor to mortality [68]. Chronic inflammation has been linked to elevated cardiovascular risk, and its treatment may improve long-term survival in HD patients [69]. A previous meta-analysis reported that, after physical training, CRP values were significantly higher, although this effect was only based on two studies with 38 patients in total [15]. Our meta-analysis involving seven trials (14 ESs) with 230 patients revealed that CRP levels were reduced following the exercise interventions. Similarly, in the sample of predominantly cardiovascular and diabetes type 2 patients, Fedewa et al. showed that participating in exercise is linked to improvements in CRP levels [70]. The authors did not find any association with age or disease vintage among the subjects, as in our regression analysis, which showed that the age of patients and time on dialysis significantly predicted the effect of an exercise intervention on CRP. In our analysis, the older the patient and the longer they remain on dialysis treatment, the weaker the effect of exercise training on lowering the CRP level.

The limitations of the included studies mostly conditioned the limitations of our meta-analysis: Outcome assessors were not blinded, intention-to-treat analyses were not included, and interventions and outcome measures were inadequately reported. As there was no attempt to identify unpublished registered clinical trials, we cannot exclude the possibility of publication bias. Some potential biases concerning the testing procedures were discovered during quality assessment. First, only twelve of the trials reported blinding of their assessors. Second, the majority of trials did not note that the patients were familiarized with the tests. Also, there is a lack of allocation concealment, and the sample size of quite a few studies (64%) was small (<20 patients in a group). Moreover, most studies did not calculate the required sample size needed for precise effect estimation.

To obtain results, patients must adhere to exercise programs. The weakness of our study is that it was not able to analyze the impact of adherence on various outcomes, and therefore we cannot be sure in what ways the specific modalities, durations, and frequencies impact patients. Exercise adherence was only reported in a few of the trials included in our meta-analysis. Therefore, it is unclear if the exercise programs were performed as planned. Our meta-analysis is not the first to show the beneficial effect of exercise for HD patients. However, our study expands on previous meta-analyses by providing more detailed assessments of the effects of different modalities of exercise training and their combinations on functional capacity, lower limb muscle strength endurance, inflammation biomarkers, and oxygen consumption. Moreover, the main strength of our study is that we conducted a meta-regression analysis and examined whether different variables moderated the effects of exercise on the selected outcomes. Similar to previous studies, regardless of the exercise modality and duration, our analysis revealed that regular exercise improves functional abilities and inflammation more than usual care.

Our data suggest that combined training offers the highest magnitude of changes in functional capacity. However, because insignificant results were observed when training modalities were compared in the sub-group analysis, this result must be taken with caution. Hence, further research on high methodological quality aimed at directly comparing the effects of combined training with aerobic or resistance training alone on physical performance among HD patients is warranted. Based on our findings, we believe that a prescription of a combined exercise program composed of aerobic and resistance exercises (three times per week for 50 min, including a warm-up and a cool-down period) in regular dialysis practice may be one of preferred options. Initial improvements in functional capacity can be expected within 12 weeks of training, with additional improvement in the following six months.

Meta-analytical evidence supports the benefits of different exercise programs for the improvement of several health-related outcomes for HD patients. However, exercise benefits for HD patients are typically reported under the assumption that the group average represents the response of most individuals. There is a wide inter-individual variability observed in human responses to the same training program. This information was confirmed by our meta-regression analysis, which showed that the age of the participants could significantly modify the effects of exercise on our outcomes. Therefore, efforts to individualize exercise prescriptions are needed in clinical practice to enhance improvements.

## 5. Conclusions

The present systematic meta-analysis review demonstrates that overall exercise programs among dialysis patients provide significant positive effects in terms of 6-min walk test performance, oxygen consumption, lower extremity strength endurance, and inflammation. Combined exercise resulted in the largest effect sizes. However, differences between this type of exercise and aerobic or resistance training alone did not reach statistical significance. The single training session duration and weekly frequency of training appear to influence the effectiveness of exercise intervention, with 50 min, three times weekly routines possibly providing the largest improvements in functional capacities. We believe that these data can help providers in exercise prescription. However, tailoring exercise interventions to the individual needs of patients remains the best approach. The present review could also be of help in planning future interventional exercise research, wherein we advise recruiting larger sample sizes, comparable interventions, and longer-term follow ups.

## Figures and Tables

**Figure 1 jcm-09-00043-f001:**
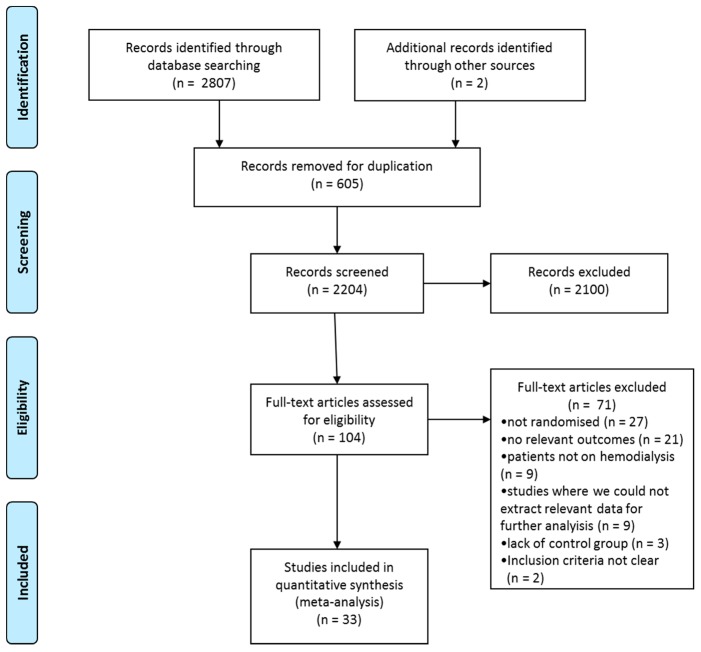
Preferred Reporting Items for Systematic Reviews and Meta-Analyses (PRISMA) flow diagram.

**Figure 2 jcm-09-00043-f002:**
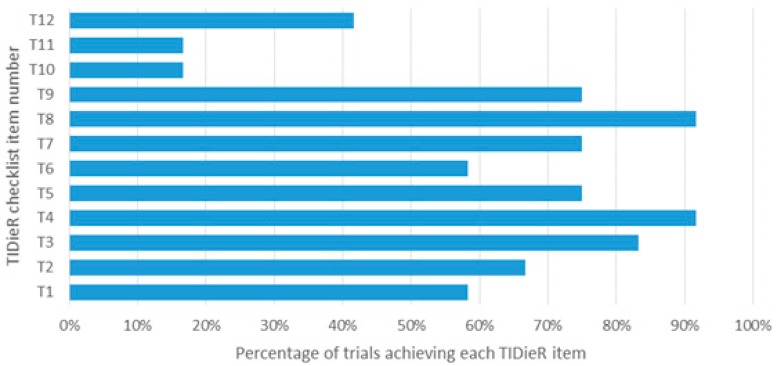
Percentage of studies achieving each TIDieR item of the experimental group.

**Figure 3 jcm-09-00043-f003:**
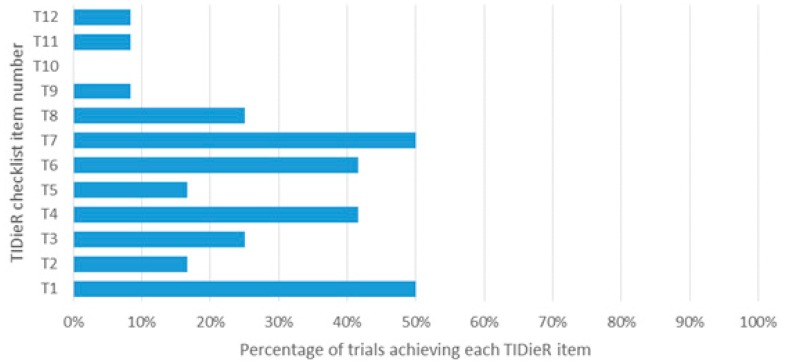
Percentage of studies achieving each TIDieR item of the control group.

**Figure 4 jcm-09-00043-f004:**
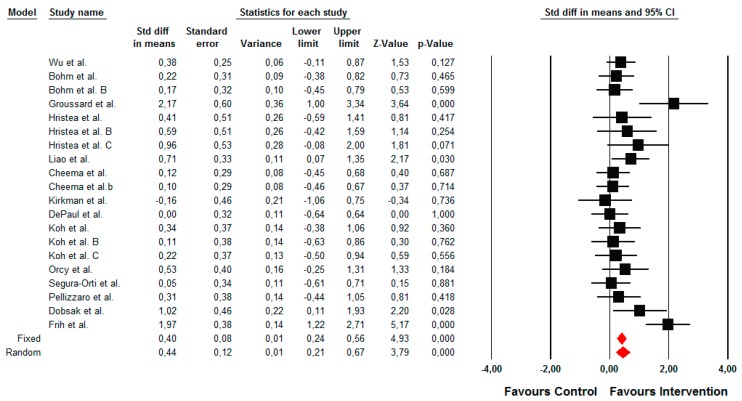
Forest plot of the standardized mean differences of the effect on the 6-min walk test when the intervention was compared with the control group.

**Figure 5 jcm-09-00043-f005:**
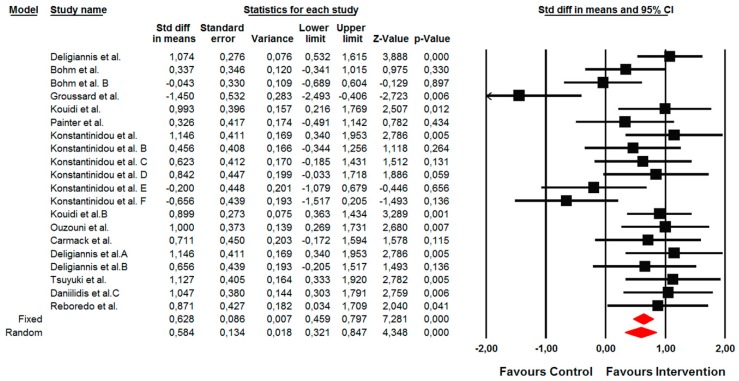
Forest plot of the standardized mean differences of the effect on Oxygen Consumption when the intervention was compared with the control group.

**Figure 6 jcm-09-00043-f006:**
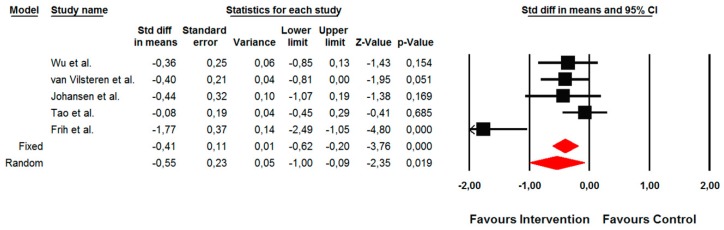
Forest plot of the standardized mean differences of the effect on a 10-repetition sit-to-stand test when the intervention was compared with the control group. Note: A negative direction of ES indicates a better effect of the intervention.

**Figure 7 jcm-09-00043-f007:**
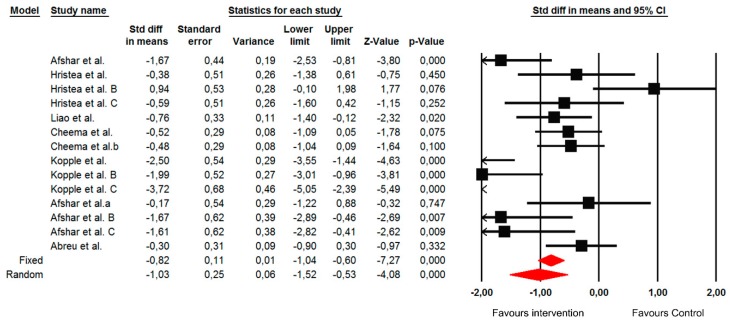
Forest plot of the standardized mean differences of the effect on the C-reactive protein when the intervention was compared with the control group. Note: A negative direction of ES indicates a better effect of the intervention.

**Table 1 jcm-09-00043-t001:** Systematic review and characteristics of the included studies selected for meta-analysis and relevant outcomes.

Study	Population Sex; Age (years) (mean ± SD)	Time on Dialysis (months)	Sample Size	Training Program	Outcome Measure	Results
Deligiannis 1999 **[39]**	M and F; Ex 48 ± 12 Con 48 ± 11	Ex 75.6 ± 36.0 Con 74.4 ± 43.2	Ex (*n* = 30) Con (*n* = 30)	Ex: AER; 6 months; 3 days/week; 90 min; 60%–70% HRmax Con: usual care	VO_2_max	Ex: 41.0% ↑ * Con: no changes
Afshar 2011 **[50]**	M; Ex 50.71 ± 21.06 Con 53 ± 19.4	Ex 25.71 ± 7.61 Con 24.86 ± 15.44	Ex (*n* = 14) Con (*n* = 14)	Ex: AER; 8 weeks; 3 days/week; 20 min; 12–15 RPE Con: usual care	CRP	Ex: 83.2% ↓ ** Con: 1.2% ↑
Wu 2014 **[28]**	M and F; Ex 45 Con 44	Ex 55.5 ± 37.3 Con 39.8 ± 29.7	Ex (*n* = 34) Con (*n* = 35)	Ex: AER; 12 weeks; 3 days/week; 13 min; 12–16 RPE Con: stretching; 12 weeks; 3 days/week; 10–15 min	6MWT STS-10	Ex: 8.09% ↓ * Con: 4.77% ↓ Ex: 5.7% ↓ * Con: 5.0% ↓
Bohm 2014 **[29]**	M and F; Ex 52 ± 14.5 Con 53 ± 16.9	Ex 37 ± 69 Con 21 ± 30	Ex (*n* = 27) Con (*n* = 26)	Ex: AER; 12 weeks; 3 days/week; 45 min; 13 RPE Con: walking program at home	peakVO_2_ 6MWT	Ex: fter 12 weeks: 9.89% ↑ after 24 weeks: no change Con: after 12 weeks: 3.31% ↓ after 24 weeks: 1.66% ↑ Ex: after 12 weeks: 6.58% ↑ after 24 weeks: 3.96% ↑ Con: after 12 weeks: 1.67% ↑ after 24 weeks: 0.05% ↓
Groussard 2015 **[31]**	M and F; Ex 66.5 ± 4.6 Con 68.4 ± 3.7	Ex 36.6 ± 8.2 Con 41.2 ± 8.1	Ex (*n* = 8) Con (*n* = 10)	Ex: AER; 3 months; 3 days/week; 30 min; 55%–60% peak power Con: usual care	6MWT peakVO_2_	Ex: 23.15% ↑ ** Con: 7.98% ↑ Ex: 2.72% ↓ Con: 14.18% ↑
Hristea 2016 **[51]**	M and F; Ex 68.5 ± 13.97 Con 70.8 ± 15.18	Ex 139 Con 96	Ex (*n* = 10) Con (*n* = 11)	Ex: AER; 6 months; 3 days/week; 30 min; 3 RPE (modified Borg scale) Con: usual care	6MWT CRP	Ex: 21.8% ↑ ** Con: 18.8% ↓ Ex: 30.6% ↓** Con: 19.8% ↓
Liao 2016 **[32]**	M and F; Ex 62 ± 8 Con 62 ± 9	Ex 71 ± 46 Con 83 ± 71	Ex (*n* = 20) Con (*n* = 20)	Ex: AER; 3 months; 3 days/week; 20 min; 13–14 RPE Con: usual care	CRP 6MWT	Ex: 37.6% ↓ * Con: 0.81% ↓ Ex: 11.1% ↑ * Con: 1.1% ↓
Cheema 2007 **[33]**	M and F; Ex 60.0 ± 15.3 Con 65.0 ± 12.9	Ex 39.6 Con 19.2	Ex (*n* = 24) Con (*n* = 25)	Ex: RT; 12 weeks; 3 days/week; 45 min; 15–17 RPE Con: usual care	6MWT CRP	Ex: 3.36% ↑ Con: 0.74% ↓ Ex: 10.26% ↓ * Con: 33.33% ↑ *
Kirkman 2014 **[34]**	M and F; Ex 48 ± 18 Con 58 ± 15	Ex 46 ± 54 Con 66 ± 47	Ex (*n* = 9) Con (*n* = 10)	Ex: RT; 12 weeks; 2 days/week; 80% of their predicted 1 RM Con: stretching	6MWT	Ex: 7.33% ↑ Con: 13.04% ↑
Kouidi 1997 [49]	M and F; Ex 49.6 ± 12.1 Con 52.8 ± 10.2	Ex 70.8 ± 58.8 Con 74.4 ± 64.8	Ex (*n* = 20) Con (*n* = 11)	Ex: COMB; 6 months; 3 days/week; 90 min; 50%–60% VO2max Con: usual care	VO_2_max	Ex: 38.09% ↑ * Con: 1.24% ↓
Painter 2002 **[52]**	M and F; Ex 55.9 ± 15.15 Con 52.8 ± 16.8	Ex 33.7 ± 35.6 Con 40.2 ± 62.4		Ex: AER; 5 months; 3 days/week; 30 min; 12–14 RPE Con: usual care	peak VO_2_	Ex: 13.3% ↑ * Con: 0.5% ↑
DePaul 2002 **[35]**	M and F; Ex 55 ± 16 Con 54 ± 14	Ex 50.4 ± 57.6 Con 55.2 ± 54.0	Ex (*n* = 20) Con (*n* = 18)	Ex: COMB; 12 weeks; 3 days/week; 30 min; 13 RPE Con: range-of-motion exercises; 12 weeks; 3 days/week; 30 min	6MWT	Ex: 0.87% ↑ Con: 0.94% ↑
Konstantinidou 2002 **[46]**	M and F; group A 46.4 ± 13.9 group B 48.3 ± 12.1 group C 51.4 ± 12.5 group D 50.2 ± 7.9	group A 78 ± 62 group B 72 ± 66 group C 62 ± 37 group D 79 ± 86	group A (*n* = 16) group B (*n* = 10) group C (*n* = 10) group D (*n* = 12)	group A: COMB; 6 months; 3 days/week; 60 min; 60%–70% HRmax group B: COMB; 6 months; 3 days/week; 70% HRmax group C: AER 6 months; 5 days/week; 50%–60% HRmax group D: usual care	peakVO_2_	group A: 42.77 % ↑ * group B: 23.93% ↑ * group C: 17.28% ↑ * group D: 3.07% ↓
van Vilsteren 2005 **[37]**	M and F; Ex 52 ± 15 Con 58 ± 16	Ex 38.6 ± 48.9 Con 46.8 ± 52.9	Ex (*n* = 53) Con (*n* = 43)	Ex: COMB; 12 weeks; 2–3 days/week; 60 min; 12–16 RPE Con: usual care	STS-10 peakVO_2_	Ex: 22.36% ↓ * Con: 0.25% ↓ Ex: 10.14% ↑ Con: 0.46% ↑
Kopple 2007 **[53]**	M and F; group 1 45.9 ± 4.1 group 2 46.0 ± 2.7 group 3 42.7 ± 3.8 group 4 41.3 ± 3.3	group 1 45.9 ± 14.1 group 2 51.9 ± 12.4 group 3 38.3 ± 5.8 group 4 51.4 ± 21.0	group 1 (*n* = 10) group 2 (*n* = 15) group 3 (*n* = 12) group 4 (*n* = 14)	Group 1: AER; 5 months; 3 days/week; 30 min; 50% peakVO_2_ Group 2: RT; 5 months; 3 days/week; 5% of the 5 RM Group 3: COMB;5 months; 3 days/week Group 4: usual care	CRP	group 1: 44.44% ↓ group 2: 20.00% ↑ group 3: 26.09% ↑ group 4: 33.33% ↑
Kouidi 2009 **[40]**	M and F; Ex 54.6 ± 8.9 Con 53.2 ± 6.1	Ex 75.6 ± 44.4 Con 74.4 ± 46.8	Ex (*n* = 30) Con (*n* = 29)	Ex: COMB; 10 months; 3 days/week; 90 min; 13 RPE Con: usual care	peakVO2	Ex: 30.49% ↑ ** Con: 1.19% ↓
Koh 2010 **[54]**	M and F; group 1 52.3 ± 10.9 group 2 52.1 ± 13.6 group 3 51.3 ± 14.4	group 1 32.1 ± 26.7 group 2 37.0 ± 31.1 group 3 25.8 ± 22.2	group 1 (*n* = 15) group 2 (*n* = 15) group 3 (*n* = 16)	group 1: AER; 6 months; 3 days/week; 30 min; 13–14 RPE group 2: walking; 6 months; 3 days/week; 30 min; 13–14 RPE group 3: usual care	6MWT	group 1: 13.61% ↑ group 2: 11.04% ↑ group 3: 4.87% ↑
Afshar 2010 **[55]**	M; group 1 50.7 ± 21.06 group 2 51 ± 16.4 group 3 53 ± 19.4	group 1 25.71 ± 7.61 group 2 24.86 ± 18.69 group 3 24.86 ± 15.44	group 1 (*n* = 7) group 2 (*n* = 7) group 3 (*n* = 7)	group 1: AER; 8 weeks; 3 days/week; 30 min; 12–16 RPE group 2: RT; 8 weeks; 3 days/week; 30 min; 15–17 RPE group 3: usual care	CRP	group 1: 83.85% ↓ ** group 2: 67.89% ↓ * group 3: 1.47% ↑
Ouzouni 2009 **[41]**	M and F; Ex 47.4 ± 15.7 Con 50.5 ± 11.7	Ex 92.4 ± 84 Con 103.2 ± 72	Ex (*n* = 19) Con (*n* = 14)	Ex: COMB; 10 months; 3 days/week; 60–90 min 13–14 RPE Con: usual care	peakVO_2_	Ex: 21.05% ↑* Con: 0.99% ↓
Orcy 2012 **[56]**	M and F; group 1 56.9 ± 14.8 group 2 55.8 ± 18.3	group 1 22.5 group 2 23	group 1 (*n* = 13) group 2 (*n* = 13)	group 1: COMB; 10 weeks; 3 days/week; 13–14 RPE group 2: RT; 10 weeks; 3 days/week; 13–14 RPE	6MWT	group 1: 9.01% ↑ * group 2: 4.48% ↓
Tao 2015 **[43]**	M and F; Ex 53.02 ± 11.62 Con 56.68 ± 9.67	Ex 83.46 ± 61.37 Con 84.70 ± 70.55	Ex (*n* = 57) Con (*n* = 56)	Ex: RT; 6 weeks; 1 day/week; 20 min; RPE 12–13 + home exercise; 1 day/week; 20 min Con: RT; 6 weeks; 1 day/week; 20 min; RPE 12–13	STS-10	Ex: 21.03% ↓ ** Con: 16.24% ↓*
Carmack 1995 **[57]**	M and F; All subjects 44.09	All subjects 29.52	Ex (*n* = 10) Con (*n* = 11)	Ex: AER; 10 weeks; 3 days/week; 25 min Con: usual care	peakVO_2_	Ex: 34.58% ↑ * Con: 9.0% ↑
Deligiannis 1999 **[39]**	M and F; group 1 46.4 ± 13.9 group 2 51.4 ± 12.5 group 3 50.2 ± 1.9	group 1 78 ± 62 group 2 62 ± 37 group 3 79 ± 86	group 1 (*n* = 16) group 2 (*n* = 10) group 3 (*n* = 12)	group 1: COMB; 6 months; 3 days/week; 90 min; 60%–70% HRmax group 2: AER; 6 months; 5 days/week; 30 min; 50%–60% HRmax + simple flexibility and muscular extension exercises group 3: usual care	VO_2_max	group 1: 42.8% ↑** group 2: 17.3% ↑* group 3: 3.07% ↓
Tsuyuki 2003 **[58]**	M and F; Ex 40.1 ± 11.9 Con 39.7 ± 10.7	Ex 25.2 Con 32.4	Ex (*n* = 17) Con (*n* = 12)	Ex: AER; 20 weeks; 2–3 days/week; 30 min; 50%–60% of the peak heart rate Con: usual care	peakVO_2_	Ex: 25.58% ↑ ** Con: 2.25% ↓
Segura-Ortí 2009 **[48]**	M and F; group 1 53.5 ± 18.0 group 2 60.1 ± 16.9	group 1 37.3 ± 34.9 group 2 53.7 ± 42.0	group 1 (*n* = 17) group 2 (*n* = 18)	group 1: RT; 6 months; 3 days/week; 25 min; 12–14 RPE group 2: AER; 6 months; 3 days/week; 25 min; 11 RPE	STS-10 6MWT	group 1: 22.31% ↓ ** group 2: 6.37% ↓ group 1: 11.21% ↑ ** group 2: 8.95% ↑
Johansen 2006 **[38]**	M and F; Ex 54.4 ± 13.6 Con 56.8 ± 13.8	Ex 33.0 Con 25.5	Ex (*n* = 20) Con (*n* = 20)	Ex: RT; 12 weeks; 3 days/week Con: usual care	STS-10	Ex: 16.11% ↓ Con: 0.66% ↓
Daniilidis 2004 **[42]**	M and F; Ex 46.7 ± 13.8 Con 53.7 ± 7.1	Ex 78 ± 62.4 Con 72 ± 81.8	Ex (*n* = 18) Con (*n* = 14)	Ex: COMB; 6 months; 3 days/week; 60 min; 75%–85% of the estimated peak heart rate Con: usual care	peakVO_2_	Ex: 42.77% ↑ * Con: 2.45% ↓
Pellizzaro 2013 **[47]**	M and F; Ex 48.9 ± 10.1 Con 51.9 ± 11.6	Ex 54 Con 54	Ex (*n* = 14) Con (*n* = 14)	Ex: RT 10 weeks; 3 days/week; 50% of 1 RM Con: usual care	6MWT	Ex: 6.84% ↑ * Con: 0.07% ↑
Dobsak 2012 **[59]**	M and F; Ex 58.2 ± 7.2 Con 60.1 ± 8.2	Ex 49.2 ± 25.2 Con 49.2 ± 27.6	Ex (*n* = 11) Con (*n* = 10)	Ex: AER; 20 weeks; 3 days/week; 20–40 min; 60% peak workload Con: usual care	6MWT	Ex: 15.02% ↑ ** Con: 3.32% ↓
Reboredo 2015 **[30]**	M and F; Ex 50.7 ± 10.7 Con 42.2 ± 13.0	Ex 39.6 ± 40.8 Con 57.6 ± 52.8	Ex (*n* = 12) Con (*n* = 12)	Ex: AER; 12 weeks; 3 days/week; 43 min; 4–6 RPE Con: usual care	peakVO_2_	Ex: 12.4% ↑ * Con: 9.2% ↓
Frih 2017 **[44]**	M; Ex 64.2 ± 3.4 Con 65.2 ± 3.1	Ex 72.7 ± 12.7 Con 73.6 ± 13.4	Ex (*n* = 21) Con (*n* = 20)	Ex: COMB; 16 weeks; 4 days/week; 40 min; 5–6 RPE (Borg 10 grade scale); 60% of 1 RMCon: usual care	6MWT STS-10	Ex: 15.94% ↑ * Con: 1.56% ↓ Ex: 16.2% ↓ * Con: 0.95% ↑
Valenzuela 2018 **[45]**	M and F; Ex 68 ± 13 Con 68 ± 11	Ex 84 ± 60 Con 60 ± 48	Ex (*n* = 27) Con (*n* = 40)	Ex: COMB; 14 weeks; 3 days/week; 50 min; RPE 13 Con: usual care	6MWT STS-10	Ex: 11.05% ↑ ** Con: 3.23% ↓ Ex: 22% ↓ ** Con: 6% ↑
Abreu 2017 **[36]**	M and F; All subjects 46.4 ± 14.6	ND	Ex (*n* = 25) Con (*n* = 19)	Ex: COMB; 12 weeks; 3 days/week; 30 min; 60% of 1 RM Con: usual care	CRP	Ex: 24.68% ↓ Con: 1.64% ↓

Abbreviations: Ex: experimental group, Con: control group, AER: aerobic training, RT: resistance training, COMB: combined training, RM: repetition maximum, CRP: C-reactive protein, STS-10: ten repetition sit-to-stand test, 6MWT: six-minute walk test, HD: hemodialysis, M: male, F: female, RPE: rate of perceived exertion, HR: heart rate, *n*: number of subjects, SD: standard deviation, ND: no data. * *p* < 0.05 indicates a significant within-group difference. ** *p* < 0.01 indicates a significant within-group difference.

**Table 2 jcm-09-00043-t002:** PEDro scale of included studies.

Study	Criterion 1	Criterion 2	Criterion 3	Criterion 4	Criterion 5	Criterion 6	Criterion 7	Criterion 8	Criterion 9	Criterion 10	Criterion 11	Score
DePaul et al. [35]	1	1	1	1	0	0	1	1	1	1	1	8
Konstantinidou et al. [46]	1	1	0	1	0	0	0	1	1	1	1	6
van Vilsteren et al. [37]	1	1	1	1	0	1	1	0	0	1	0	6
Johansen et al. [38]	1	1	0	1	0	1	1	1	0	1	1	7
Cheema et al. [33]	1	1	1	1	0	0	0	1	1	1	1	7
Wu et al. [28]	1	1	1	1	0	0	0	1	0	1	1	6
Bohm et al. [29]	1	1	1	1	0	0	0	0	1	1	1	6
Groussard et al. [31]	1	1	0	1	0	0	0	1	1	1	1	6
Hristea et al. [51]	1	1	1	1	0	0	0	0	1	1	1	6
Liao et al. [32]	1	1	0	1	0	0	0	1	1	1	1	6
Kirkman et al. [34]	1	1	0	1	0	0	1	0	1	1	1	6
Koh et al. [54]	1	1	1	1	0	0	0	0	0	1	1	5
Orcy et al. [56]	1	1	1	1	0	0	1	1	1	1	1	8
Segura-Orti et al. [48]	1	1	1	1	1	0	1	1	1	1	1	9
Pellizzaro et al. [47]	1	1	1	1	0	0	0	1	0	1	1	6
Dobsak et al. [59]	1	1	0	1	0	0	0	1	1	1	1	6
Frih et al. [44]	1	1	1	1	0	0	1	0	0	1	1	6
Valenzuela et al. [45]	1	0	0	1	1	0	0	1	1	1	1	6
Afshar et al. [50]	1	1	0	1	0	0	0	0	0	1	1	4
Kopple et al. [53]	1	1	0	1	0	0	0	0	0	1	1	4
Afshar et al. [55]	1	1	0	1	0	0	0	0	0	1	1	4
Abreu et al. [36]	1	1	0	1	0	0	0	0	0	1	1	4
Tao et al. [43]	1	1	1	1	0	0	1	1	1	1	1	8
Deligiannis et al. [60]	1	1	0	1	0	0	0	1	0	1	0	4
Kouidi et al. [49]	1	1	0	1	0	0	0	1	0	1	1	5
Painter et al. [52]	1	1	0	1	0	0	0	0	1	1	1	5
Kouidi et al. [40]	1	1	0	1	0	0	1	1	0	1	1	6
Ouzouni et al. [41]	1	1	0	1	0	0	0	1	0	1	1	5
Carmack et al. [57]	1	1	0	1	0	0	0	0	0	1	0	3
Deligiannis et al. [39]	1	1	0	1	0	0	0	0	0	1	0	3
Tsuyuki et al. [58]	1	1	0	1	0	0	0	1	1	1	0	5
Daniilidis et al. [42]	1	1	0	1	0	0	0	1	0	1	0	4
Reboredo et al. [30]	1	1	0	1	0	0	0	1	0	1	1	5

**Table 3 jcm-09-00043-t003:** Meta-regression for training the variables of different subscales to predict the intervention effect on measures of functional fitness and inflammation biomarkers.

	Coefficient	Standard Error	95% Lower CI	95% Upper CI	*z* Value	*p* Value
**6-min walk test**						
Age of patients	0.034	0.015	0.004	0.063	2.243	0.025 *
Time on dialysis	0.005	0.003	−0.002	0.012	1.382	0.167
Duration of intervention (weeks)	−0.011	0.019	−0.048	0.026	−0.576	0.556
Weekly frequency	1.631	0.398	0.852	2.410	4.102	<0.001 **
Duration of single training session	0.006	0.010	−0.014	0.026	0.561	0.575
Total number of training sessions	0.000	0.006	−0.012	0.013	0.013	0.989
Total training volume	0.000	0.000	0.000	0.000	0.319	0.750
**Oxygen consumption**						
Age of patients	−0.013	0.013	−0.039	0.013	−0.970	0.332
Time on dialysis	0.011	0.006	−0.001	0.023	1.760	0.078
Duration of intervention (weeks)	0.024	0.017	−0.009	0.057	1.447	0.148
Weekly frequency	−0.322	0.227	−0.768	0.124	−1.417	0.157
Duration of single training session	0.017	0.008	0.002	0.032	2.248	0.025 *
Total number of training sessions	0.003	0.005	−0.007	0.012	0.520	0.603
Total training volume	0.000	0.000	0.000	0.000	1.730	0.084
**C-reactive protein**						
Age of patients	0.077	0.024	0.031	0.124	3.272	0.001 **
Time on dialysis	0.013	0.006	0.001	0.024	2.113	0.035 *
Duration of intervention (weeks)	−0.035	0.043	−0.119	0.049	−0.822	0.411
Weekly frequency	NA	NA	NA	NA	NA	NA
Duration of single training session	0.000	0.036	−0.071	0.071	−0.001	0.999
Total number of training sessions	−0.012	0.014	−0.040	0.016	−0.822	0.411
Total training volume	0.000	0.000	−0.001	0.000	−0.654	0.513

NA—not applicable; * *p* < 0.05; ** *p* < 0.01.

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
