# Peer review of "Exercise-Based Interventions in Hemodialysis Patients: A Systematic Review with a Meta-Analysis of Randomized Controlled Trials"

_jcm, 2019, doi:10.3390/jcm9010043_

Round 1

Reviewer 1 Report

The authors set out to examine the benefits of exercise in hemodialysis patients through examining the available literature with a meta-analysis study. The question explored is pertinent and the importance of the topic is relatively high.

In the abstract, lines 21, 22, 23 describe non statistically significant findings. Suggest to add the phrase not statistically significant.

The language is a bit complex and would benefit from simplification at the expense of some of the detailed statistical terms that may not be of benefit. Suggest to simplify the language and make the manuscript a bit more brief by eliminating the few repetitions of ideas in discussion.

Line 73 needs simplified for an easier read.

Figure 1 The box of "records after duplicates removed (n=605)" can be better labelled as "records removed for duplication" .

Line 337 needs clarification. There seems to be an extra preposition "of the"?

I find the sentences that include terms in succession like "possibly small, likely moderate" a bit hard to follow. It may be simplified leaving the numbers speak for themselves.

I agree with the authors strengths and weaknesses reported.

The conclusion is a bit verbose for delivering the nutshell of the data e.g. the sentence: We believe... in line 398 can be replaced simply but these data can help providers tailor exercise interventions to the individual needs of their HD patients.

In the end, I commend the authors on the complex work and attention to details, although as outlined not every detail should be reported especially when the data are tabulated for easy perusal.

Author Response

The authors set out to examine the benefits of exercise in hemodialysis patients through examining the available literature with a meta-analysis study. The question explored is pertinent and the importance of the topic is relatively high.

Our response: We are grateful for your detailed review of our manuscript and for providing some insightful and thought-provoking suggestions to strengthen our manuscript. We feel we have sufficient responses to each of your major concerns, which are detailed below. We hope that they alleviate the concerns you have and believe that you will find our manuscript acceptable. Our changes in the manuscript are highlighted in yellow color. After our revision, we sent a manuscript to English editing service; that is why we could not keep our track changes and only kept theirs. We also have our version before the English editing with track changes if you would need it.

Point 1: In the abstract, lines 21, 22, 23 describe non statistically significant findings. Suggest to add the phrase not statistically significant.

Response 1: Thank you for your suggestion. We have added the requested phrase (line 20).

Point 2: The language is a bit complex and would benefit from simplification at the expense of some of the detailed statistical terms that may not be of benefit. Suggest to simplify the language and make the manuscript a bit more brief by eliminating the few repetitions of ideas in discussion.

Response 2: We have removed the detailed statistical terms. The results section is now simplified and hopefully easier to understand. Please observe also that discussion was revised, shortened and verified for the omission of repetitive ideas. Additionally, we sent our manuscript to the English editing service.

Point 3: Line 73 needs simplified for an easier read.

Response 3: Thank you, we have changed the sentence. It has been revised to the following statement (line 71-74): “Accordingly, an evidence-based answer to the uncertainty over the most beneficial type of exercise for dialysis patients is needed. Thus, we conducted this meta-analysis to answer this question in a more inclusive way with a more complete inclusion of relevant studies than previous meta-analyses”.

Point 4: Figure 1 The box of "records after duplicates removed (n=605)" can be better labelled as "records removed for duplication".

Response 4: We have changed the figure box according to your suggestion.

Point 5: Line 337 needs clarification. There seems to be an extra preposition "of the"?

Response 5: We agree with the remark; the sentence was amended accordingly. Thank you for noticing the unnecessary preposition (line 331). The whole sentance now goes: »Our results showed that HD patients could improve their performance in 10 repetition sit-to-stand test with no significant difference among the exercise modalities used.«

Point 6: I find the sentences that include terms in succession like "possibly small, likely moderate" a bit hard to follow. It may be simplified leaving the numbers speak for themselves.

Response 6: Thank you for this remark. We agree that it was hard to follow; therefore, we removed the suggested terms (abstract, results, discussion).

Point 7: I agree with the authors strengths and weaknesses reported.

Response 7: Thank you for this comment.

Point 8: The conclusion is a bit verbose for delivering the nutshell of the data e.g. the sentence: We believe... in line 398 can be replaced simply but these data can help providers tailor exercise interventions to the individual needs of their HD patients.

Response 8: Thank you for your comment; the conclusion was modified to the following (line 388-399): »The present systematic meta-analysis review demonstrates that overall exercise programs among dialysis patients provide significant positive effects in terms of 6-minute walk test performance, oxygen consumption, lower extremity strength endurance, and inflammation. Combined exercise resulted in largest effect sizes; however, differences between this type of exercise and aerobic or resistance training alone did not reach statistical significance. The single training session duration and weekly frequency of training appear to influence the effectiveness of exercise intervention, with 50 minutes, three times weekly routines possibly providing the largest improvements in functional capacities. We believe that these data can help providers in exercise prescription. However, tailoring exercise interventions to the individual needs of patients remains the best approach. The present review could also be of help in planning future interventional exercise research, wherein we advise recruiting larger sample sizes, comparable interventions, and longer-term follow ups.«

Please see also answers to reviewer 2.

In the end, I commend the authors on the complex work and attention to details, although as outlined not every detail should be reported especially when the data are tabulated for easy perusal.

Our response: We are very thankful for this comment. We appreciate your efforts to improve this work. We feel that several important points in our submission were identified, and as a result of these revisions, the manuscript is now much stronger and clearer.

Reviewer 2 Report

JCM-666950  Review

Exercise-based Interventions in Hemodialysis Patients: A Systematic review with Meta-analysis of Randomized Controlled Trials

In this meta-analysis different exercise interventions have been compared in hemodialysis patients. While the topic is very important and the manuscript is generally well-written, several concerns may be considered.

The main concern is that the conclusions drawn by the Authors seem different from the results obtained. Both in the abstract and conclusions the Authors state that “Results have shown largest beneficial effect sizes for combined interventions compared to solely aerobic or resistance training at physical performance outcomes…”. In fact, no statistical differences have been shown in the results: p=0.094 for 6-MWT, p=0,522 for VO2max, p=0.786 for 10 repetition sit-to-stand test, and p=0.102 for CRP. Therefore, while overall benefit of exercise for hemodialysis patients was confirmed in this meta-analysis as in several previous reviews, available data seem not sufficient to state potential differences between different exercise modes. 

It is also contradictory to the statements in the discussion:

Lines 299-300: “Here, combined aerobic and resistance training provided the largest beneficial effect sizes…”

Lines 376-377: “Our current meta-analysis of randomized controlled studies provides a new aspect by showing that HD patients benefit most from the combined training modalities.”

Other comments

Data shown in the abstract present ES for 6-MWT and VO2max as compared with inactive controls, not controls; this should be clearly stated or corrected.

Line 230; should be probably: p=0.498

Lines 280-281: what about combined exercise – no data has been given.

While the selection of the three performance measures to assess the efficacy of different exercise interventions seem justified, the rationale for additional inclusion of only CRP is not clear. Why not to include quality of life or depression measures?

Author Response

In this meta-analysis different exercise interventions have been compared in hemodialysis patients. While the topic is very important and the manuscript is generally well-written, several concerns may be considered.

Our comment: We are grateful for your careful evaluation and helpful comments to our manuscript. We have carefully considered your comments in preparing our revision, which has resulted in a paper that is clearer, broader and more compelling. After our revision, we sent a manuscript to English editing service; that is why we could not keep our track changes and only kept theirs. We also have our version before the English editing with track changes if you would need it. Our changes in the manuscript are highlighted in yellow color.

Point 1: The main concern is that the conclusions drawn by the Authors seem different from the results obtained. Both in the abstract and conclusions the Authors state that “Results have shown largest beneficial effect sizes for combined interventions compared to solely aerobic or resistance training at physical performance outcomes…”. In fact, no statistical differences have been shown in the results: p=0.094 for 6-MWT, p=0,522 for VO2max, p=0.786 for 10 repetition sit-to-stand test, and p=0.102 for CRP. Therefore, while overall benefit of exercise for hemodialysis patients was confirmed in this meta-analysis as in several previous reviews, available data seem not sufficient to state potential differences between different exercise modes.

It is also contradictory to the statements in the discussion:

Lines 299-300: “Here, combined aerobic and resistance training provided the largest beneficial effect sizes…”

Lines 376-377: “Our current meta-analysis of randomized controlled studies provides a new aspect by showing that HD patients benefit most from the combined training modalities.”

Response 1: Thank you for pointing out this concern. We clarified our findings in the abstract, discussion and conclusion section. Combined type of training did not show any statistical differences but it had the largest effects: ES=0.71 for 6MWT, ES=0.67 for VO2max, and ES= -0.69 for 10 repetition sit-to-stand test. There were no studies found which investigated the impact of combined training on CRP (line 278). Our data suggest that combined training offers the highest magnitude of change in functional capacity. However, due to insignificant results were observed when training modalities were compared in sub-group analysis, current findings must be taken with caution. In hence, more experimental research with the high methodological quality aiming to directly compare the effects of combined training modality with aerobic or resistance training alone, on physical performance in HD patients is warranted.

So we have rewritten the abstract to the following (line 25-28): »Overall, the results showed the numerically largest effect sizes for combined types compared to solely aerobic or resistance training types, with the differences between training types not reaching statistical significance.«

As it goes for the mentioned lines 299-300, we agree that it sounds contradictory. The phrase »combined aerobic and resistance training« actually means combined training (a combination of aerobic and resistance training). We have now changed this phrase with the phrase »combined training« only. We apologize for this confusion and are grateful for this remark.

Finally, lines 367-77 were replaced by the following statement (line 369-376): Our data suggest that combined training offers the highest magnitude of changes in functional capacity. However, because insignificant results were observed when training modalities were compared in the sub-group analysis, this result must be taken with caution. Hence, further research on high methodological quality aimed at directly comparing the effects of combined training with aerobic or resistance training alone on physical performance among HD patients is warranted. Based on our findings, we believe that a prescription of a combined exercise program composed of aerobic and resistance exercises (3 times per week for 50 minutes, including a warm-up and a cool-down period) in regular dialysis practice may be one of preferred options.

Point 2: Data shown in the abstract present ES for 6-MWT and VO2max as compared with inactive controls, not controls; this should be clearly stated or corrected.

Response 2: We apologize for this omission and have now moved the mentioned statement to the comparison between the intervention group and inactive controls (line 18).

Point 3: Line 230; should be probably: p=0.498

Response 3: Thank you for identifying this oversight (line 225).

Point 4: Lines 280-281: what about combined exercise – no data has been given.

Response 4: We value your suggestions. However, we did not find any studies that investigated the effects of combined exercise on CRP. See line 278, we added: “We found no studies investigating the effect of combined training on CRP.”

Point 5: While the selection of the three performance measures to assess the efficacy of different exercise interventions seem justified, the rationale for additional inclusion of only CRP is not clear. Why not to include quality of life or depression measures?

Response 5: Thank you for pointing out this gap in our design. We are well aware of the importance of quality of life and depression in HD patients. However, we have stick to the objective measures in our study. Moreover, the recent meta-analysis[1,2] included quality of life. We did not want to replicate the results.

Reference

Young, H.M.L.; March, D.S.; Graham-Brown, M.P.M.; Jones, A.W.; Curtis, F.; Grantham, C.S.; Churchward, D.R.; Highton, P.; Smith, A.C.; Singh, S.J.; et al. Effects of intradialytic cycling exercise on exercise capacity, quality of life, physical function and cardiovascular measures in adult haemodialysis patients: A systematic review and meta-analysis. Nephrol. Dial. Transplant. 2018, 33, 1436–1445. Gomes Neto, M.; de Lacerda, F.F.R.; Lopes, A.A.; Martinez, B.P.; Saquetto, M.B. Intradialytic exercise training modalities on physical functioning and health-related quality of life in patients undergoing maintenance hemodialysis: systematic review and meta-analysis. Clin. Rehabil. 2018, 32, 1189–1202.

Round 2

Reviewer 1 Report

Agree with changes and appropriate prompt response 

Reviewer 2 Report

Authors have appropriately responded to reviewer's comments